# From Machine Learning to Patient Outcomes: A Comprehensive Review of AI in Pancreatic Cancer

**DOI:** 10.3390/diagnostics14020174

**Published:** 2024-01-12

**Authors:** Satvik Tripathi, Azadeh Tabari, Arian Mansur, Harika Dabbara, Christopher P. Bridge, Dania Daye

**Affiliations:** 1Department of Radiology, Massachusetts General Hospital, Boston, MA 02114, USA; stripathi3@mgh.harvard.edu (S.T.); atabari@mgh.harvard.edu (A.T.); arianmansur@hms.harvard.edu (A.M.); cbridge@mgh.harvard.edu (C.P.B.); 2Athinoula A. Martinos Center for Biomedical Imaging, Charlestown, MA 02129, USA; 3Harvard Medical School, Boston, MA 02115, USA; 4Boston University Chobanian & Avedisian School of Medicine, Boston, MA 02118, USA; hdabbara@bu.edu

**Keywords:** artificial intelligence (AI), artificial neural network (ANN), future perspectives, machine learning (ML), pancreatic adenocarcinoma (PAC), review

## Abstract

Pancreatic cancer is a highly aggressive and difficult-to-detect cancer with a poor prognosis. Late diagnosis is common due to a lack of early symptoms, specific markers, and the challenging location of the pancreas. Imaging technologies have improved diagnosis, but there is still room for improvement in standardizing guidelines. Biopsies and histopathological analysis are challenging due to tumor heterogeneity. Artificial Intelligence (AI) revolutionizes healthcare by improving diagnosis, treatment, and patient care. AI algorithms can analyze medical images with precision, aiding in early disease detection. AI also plays a role in personalized medicine by analyzing patient data to tailor treatment plans. It streamlines administrative tasks, such as medical coding and documentation, and provides patient assistance through AI chatbots. However, challenges include data privacy, security, and ethical considerations. This review article focuses on the potential of AI in transforming pancreatic cancer care, offering improved diagnostics, personalized treatments, and operational efficiency, leading to better patient outcomes.

## 1. Introduction

While pancreatic cancer is the 12th most common cancer worldwide, it remains one of the most aggressive cancers with very poor prognosis and increasing incidence [1]. In 2020, there were 495,773 new cases of pancreatic cancer worldwide with almost as many deaths (466,003) [2]. One of the greatest challenges with pancreatic cancer is that it is often diagnosed very late, with approximately 80% of patients diagnosed with locally advanced or distant metastatic disease [3]. The reasons for this challenge are multifaceted and include the lack of distinctive clinical symptoms early on, the lack of specific molecular markers, and the pancreas’s deep-seated retro-peritoneal location that is surrounded by complex structures [4]. Moreover, the pancreas resides in a highly vascularized environment, facilitating rapid cancer metastasis and contributing to the exceptionally aggressive nature of the disease [4]. Despite these challenges, advances in imaging technology—through the use of computed tomography (CT), magnetic resonance (MR), 18fluoro-2-deoxy-d-glucose positron emission tomography/computed tomography (18FDG PET/CT), and endoscopic ultrasound (EUS)—have allowed for the better diagnosis and management of pancreatic cancer [5]. However, there is still room for improvement, given the heterogeneity and lack of consensus in international guidelines for image-based stratification and treatment response prediction [5,6]. Furthermore, biopsies are still oftentimes required to confirm pancreatic cancer. Still, histopathology analysis is also challenging due to the significant morphological heterogeneity in tumors and the limited amount of tissue that is able to be collected [5,7,8,9]. Like imaging features, histopathological evaluation of response to treatment is still not precise [10]. 

Artificial intelligence (AI) is revolutionizing the healthcare industry by offering innovative solutions to complex challenges. AI algorithms are programmed to analyze and interpret vast amounts of data, make informed decisions, and learn from their experiences [11]. In healthcare, AI is being leveraged to improve diagnosis, treatment, and patient care.

One of the significant contributions of AI in healthcare is its ability to enhance diagnostic accuracy. AI algorithms can analyze medical images, such as X-rays, MRIs, and CT scans, with remarkable precision, assisting healthcare professionals in detecting diseases at their early stages, shown in Figure 1. AI-powered diagnostic systems can swiftly process and compare data from a wide range of cases, providing physicians with valuable insights and supporting them in making accurate diagnoses [12,13].

Moreover, AI is playing a crucial role in personalized medicine. By analyzing large datasets that include patient health records, genetic information, and treatment outcomes, AI algorithms can identify patterns and correlations that help tailor treatment plans to individual patients. This enables healthcare providers to deliver targeted therapies, predict disease progression, and reduce adverse effects [14].

Additionally, AI is streamlining administrative tasks and improving operational efficiency in healthcare facilities. Natural language processing (NLP) algorithms can automate tasks like medical coding and documentation, reducing the burden on healthcare professionals and minimizing errors. AI chatbots are being used to provide patients with round-the-clock assistance, answer their queries, schedule appointments, and even provide basic medical advice [15].

Despite its significant benefits, the adoption of AI in healthcare also presents challenges. Ensuring data privacy and security is paramount, as healthcare data are highly sensitive and must be protected. Additionally, ethical considerations surrounding AI in healthcare, such as transparency, accountability, and bias mitigation, require careful attention to ensure fair and responsible deployment [16,17].

In this article, we discuss the applications and potential of AI holds in transforming pancreatic cancer care by improving diagnostics, enabling personalized treatments, and enhancing operational efficiency, illustrated in Figure 2. With continued advancements and responsible implementation, AI has the potential to revolutionize the way healthcare is delivered, leading to improved patient outcomes and a more efficient healthcare system overall.

## 2. AI Techniques

The successful application of artificial intelligence (AI) in identifying individuals at high risk for pancreatic cancer opens avenues for a nuanced discussion on the broader implications and potential applications of AI in pancreatic cancer care [4]. Beyond the remarkable achievement of early risk prediction, exploring the specific AI techniques and models employed in diagnostics and personalized medicine becomes imperative. The primarily utilizes AI algorithms trained on extensive patient records to predict pancreatic cancer risk, showcasing the potential for population-wide screening [18,19]. However, a more in-depth examination of the underlying AI methodologies, such as machine learning or deep learning techniques, could provide insights into the robustness and adaptability of these models in handling diverse datasets [20,21]. Furthermore, delving into the prospects of integrating AI into personalized medicine approaches for pancreatic cancer would enrich the discussion. AI has the potential to tailor diagnostic and treatment strategies based on individual patient profiles, optimizing clinical decision making and potentially improving outcomes [22,23,24]. Discussing the scalability of these AI applications in diverse healthcare settings, the need for standardized protocols, and the integration of AI into existing clinical workflows would contribute to a comprehensive understanding of how AI could reshape the landscape of pancreatic cancer care [25]. Moreover, considering the evolving nature of AI technologies, it becomes crucial to address the ethical implications, regulatory considerations, and ongoing efforts in refining these models for real-world clinical applications [17,26,27]. Overall, a more detailed exploration of the specific AI techniques employed and their potential applications in diagnostics and personalized medicine is provided in this section to enhance the discussion surrounding the transformative role of AI in pancreatic cancer care.

### 2.1. Machine Learning

Machine learning (ML) involves training algorithms on large datasets to learn patterns and make predictions. In the field of pancreatic cancer research, ML techniques have been applied to various tasks, including diagnosis, prognosis, treatment prediction, and response prediction [28].

Support vector machines (SVMs) are a popular ML algorithm that have been employed for pancreatic cancer diagnosis. SVMs can learn to classify pancreatic tumors based on features extracted from medical imaging data, such as CT scans or MRI images. By training on labeled datasets, SVM models can distinguish between healthy pancreatic tissue and cancerous lesions, aiding in the accurate detection and diagnosis of pancreatic cancer [29].

Random forests are another supervised learning algorithm that have been successfully utilized in pancreatic cancer research. Random forest models are capable of handling high-dimensional datasets and can incorporate various clinical, molecular, and demographic features to predict patient outcomes, such as survival rates or treatment response. These models have the potential to assist clinicians in making more informed decisions about treatment strategies and patient management [30].

Unsupervised learning techniques, such as clustering and dimensionality reduction, are valuable for identifying subtypes or patterns within pancreatic cancer data. Clustering algorithms, such as k-means or hierarchical clustering, group similar patient profiles based on their clinical or molecular characteristics [31,32]. These clusters can provide insights into distinct subgroups of pancreatic cancer patients, which may have implications for personalized treatment approaches or targeted therapies.

Dimensionality reduction techniques, such as principal component analysis (PCA) or t-SNE (t-distributed stochastic neighbor embedding), are utilized to reduce the dimensionality of high-dimensional datasets while preserving important information [33,34,35]. By visualizing the reduced dataset, researchers can identify patterns or clusters that may not be immediately apparent in the original data. This can aid in uncovering hidden relationships or identifying novel features relevant to pancreatic cancer [36,37,38].

ML techniques have also been applied to predict treatment response and drug efficacy in pancreatic cancer. By training ML models on genomic, proteomic, or transcriptomic data, researchers can identify molecular signatures associated with treatment response or resistance. These models can help predict which patients are likely to benefit from specific therapies, allowing for more personalized treatment approaches and avoiding unnecessary interventions [39,40].

Overall, ML techniques, including supervised learning algorithms like support vector machines and random forests, as well as unsupervised learning techniques like clustering and dimensionality reduction, are very valuable in pancreatic cancer research. They enable researchers to extract meaningful insights from complex datasets, improve diagnostic accuracy, predict patient outcomes, and facilitate personalized treatment strategies.

### 2.2. Deep Learning

Deep learning is a specific area within the broader field of machine learning that utilizes artificial neural networks consisting of multiple layers to effectively analyze intricate data and derive sophisticated features. The utilization of convolutional neural networks (CNNs) has been extensively employed in the realm of image analysis, specifically in the identification and partitioning of pancreatic cancer from medical images such as CT scans or MRI [41,42,43,44]. Recurrent neural networks (RNNs) have demonstrated their applicability in the analysis of time-series data, particularly in the context of predicting patient outcomes or monitoring treatment response in pancreatic cancer [42,45,46]. A trend analysis of AI in pancreatic cancer research is provided in Figure 3.

#### 2.2.1. Natural Language Processing

The field of natural language processing (NLP) pertains to a subdomain of AI that is dedicated to comprehending and manipulating human language. NLP techniques have played a pivotal role in the domain of pancreatic cancer research by facilitating the extraction of valuable insights from diverse sources such as electronic health records (EHRs), clinical reports, and the biomedical literature [47,48,49].

The process of identifying and extracting particular entities from textual data, such as medical terminologies, genes, proteins, or drug nomenclatures, is referred to as named entity recognition [50,51]. NLP techniques have the potential to facilitate the identification of pertinent entities, such as pancreatic tumors, biomarkers, or treatment modalities, from clinical reports or research articles in the domain of pancreatic cancer. This facilitates the efficient collection of vital data by researchers, allowing for the creation of organized databases that can be subjected to subsequent analysis.

The application of text mining in pancreatic cancer research is a significant technique within the field of NLP. Text mining is a process that entails the automated analysis and extraction of valuable insights from vast amounts of unstructured textual data. Through the utilization of text-mining methodologies in the biomedical literature, scholars can reveal latent knowledge and identify innovative associations among various factors associated with pancreatic cancer. NLP has the potential to assist in the identification of significant genes or molecular pathways that are linked to the development, progression, or treatment response of pancreatic cancer [52,53].

The application of NLP methodologies to EHRs has demonstrated significant efficacy in the domain of pancreatic cancer investigation. NLP facilitates the execution of extensive retrospective studies by extracting pertinent data from EHRs, including patient demographics, medical history, treatment particulars, and clinical outcomes [54,55,56,57]. The studies mentioned above have the potential to offer valuable insights regarding the efficacy of particular treatments, patient prognoses, and determinants that impact the advancement of pancreatic cancer [58,59].

The utilization of NLP techniques enables researchers to extract valuable information from a variety of sources, which can subsequently facilitate the identification of insights that may prove useful in the realms of diagnosis, treatment decision making, and research prioritization [55,60,61]. NLP has the potential to aid in the detection of particular clinical features or trends that signify the presence of pancreatic cancer. This can also facilitate comprehension of the efficacy of various treatment modalities and their corresponding results. NLP facilitates the process of gathering and synthesizing large amounts of data by automating the extraction and analysis of information [62,63]. This leads to more informed decision making and improved patient care, as researchers are able to process and analyze data efficiently.

NLP facilitates the retrieval of significant data from electronic health records, clinical reports, and the biomedical literature. This, in turn, offers researchers a more profound comprehension of the disease and reinforces decision making based on evidence.

#### 2.2.2. Computer Vision

The application of computer vision techniques involves the use of algorithms for image analysis and processing to derive significant insights from medical images, including but not limited to computed tomography (CT) scans, magnetic resonance imaging (MRI), or histopathological slides [14,64,65,66]. Computer vision has been utilized in diverse tasks within the domain of pancreatic cancer research, such as image segmentation, lesion detection and localization, classification and characterization, image registration and fusion, and quantitative analysis [67].

It can be used to enhance the process of dividing an image into distinct regions or objects of interest, commonly known as image segmentation. Image segmentation techniques are employed within the domain of pancreatic cancer to accurately demarcate the perimeters of pancreatic tumors or other pertinent anatomical structures. Precisely delineating the neoplastic area enables scholars to measure its dimensions, morphology, and capacity, furnishing significant insights for the purpose of identification, therapeutic strategizing, and tracking the advancement of the ailment [9,49,68,69,70].

We can also utilize CNNs for lesion detection and localization for automated identification of anomalous or dubious regions in medical imagery. The application of computer vision techniques in pancreatic cancer research has the potential to facilitate the identification and localization of pancreatic tumors and other lesions. Through the automated identification of these regions, medical professionals can concentrate their efforts on the specific areas of concern, thereby enabling enhanced precision in diagnosis and treatment strategizing [44,70,71,72,73]. These algorithms can also help classify tumors into distinct subtypes or determine their malignancy by extracting pertinent features from medical images, such as texture, shape, or intensity patterns [7,74,75]. These extracted data hold significant value in terms of prognostication, informing treatment choices, and forecasting patient results.

### 2.3. Transfer Learning

The application of transfer learning involves utilizing acquired knowledge from a specific task to enhance the performance of a related task [76,77]. The process of transferring knowledge and insights gained from solving a particular problem to solve a related but distinct problem effectively is facilitated by this approach. The utilization of transfer learning has garnered considerable interest in the realm of pancreatic cancer investigation owing to its capacity to surmount data constraints and enhance prognostic models [78,79].

One way is to use pre-trained models, which are neural networks that have undergone training on extensive datasets, often sourced from diverse domains or tasks [80,81]. The models have acquired universal features and patterns that may have relevance to multiple tasks [82]. Pre-trained models, such as convolutional neural networks (CNNs) that have been trained on general medical images or other cancer types, can be employed by researchers as a foundation for training their own pancreatic cancer-specific models [83]. Through the process of fine-tuning pre-existing models using pancreatic cancer data, the models can acquire a deeper understanding of the distinctive characteristics and features that are specific to pancreatic cancer. This can result in enhanced performance, even when the amount of available data is limited [84].

The concept of transfer learning involves the transfer of knowledge across various levels of a model, in addition to the utilization of pre-trained models, which is called knowledge transfer [69,85,86,87]. As an illustration, researchers may impart knowledge through the utilization of the pre-trained model’s lower-level layers as feature extractors, followed by the incorporation of supplementary layers that are tailored to the pancreatic cancer assignment. By leveraging the pre-trained model’s generic features, the proposed approach enables the model to adapt to the specific features of pancreatic cancer.

Lastly, the utilization of transfer learning facilitates the process of domain adaptation [88,89,90], whereby models that have been trained on a particular domain can be adjusted to achieve optimal performance on a distinct domain. In the realm of pancreatic cancer, a plausible approach would be to develop a model by utilizing data from analogous cancer types or other biomedical fields, followed by refining the model through the utilization of accessible pancreatic cancer data. By enabling the model to capture cancer-related features that possess transferability to pancreatic cancer, its performance is enhanced.

## 3. Application of AI in Clinical Studies

Various AI models with huge and complex architectures are being applied to medical imaging and oncology to improve diagnostics, decision making, and care [27,91,92]. AI algorithms analyze vast amounts of patient data to assist medical professionals in making more informed decisions about care, outperforming traditional tools like the modified early warning score (MEWS) commonly used by hospitals to calculate the risk for clinical deterioration in a patient over the next several hours [93]. The greatest application of AI in diagnostics so far has been in imaging. AI’s ability to recognize and process a great amount of both structured and unstructured data has led to nearly 400 Food and Drug Administration approvals of AI algorithms for the radiology field [94]. AI tools are helping radiologists process large volumes of imaging data with advanced image reconstruction algorithms, while helping to improve the consistency and accuracy of medical imaging and diagnostics [94]. AI models have also been used to predict the risk of cancer recurrence and to identify the most effective treatment options for individual patients [95]. In this section, we discuss the application of various AI techniques with different clinical use cases for pancreatic cancer. 

### 3.1. Early Detection 

The early detection of pancreatic cancer is a critical factor in improving patient outcomes, as it is often diagnosed at an advanced stage when treatment options are limited [19]. AI has the potential to aid in the early detection of pancreatic cancer by analyzing medical data and identifying patterns that may indicate the presence of the disease. Deep learning techniques can be trained on large datasets to accurately identify early stage pancreatic cancer based on characteristic imaging features or use morphology features to build segmentation frameworks for the pancreas [96,97,98]. AI algorithms can integrate various patient data, such as age, family history, lifestyle factors, and medical history, to detect an individual’s developing pancreatic cancer early. AI can also analyze a patient’s electronic health records, including medical history, laboratory results, and diagnostic reports, to identify potential indicators of pancreatic cancer [99,100,101]. By processing and interpreting vast amounts of data, AI algorithms can detect subtle patterns and abnormalities that may go unnoticed by clinicians.

### 3.2. Diagnosis and Classification 

Pancreatic cancer is one of the most lethal malignancies, with a five-year survival rate of less than 10% [102]. Early detection and accurate diagnosis are crucial for improving the prognosis and treatment outcomes of patients. However, pancreatic cancer is often asymptomatic in the early stages and difficult to diagnose using conventional imaging techniques. Moreover, pancreatic cancer has a high degree of heterogeneity and can be classified into different subtypes with distinct molecular features and clinical implications [103]. Therefore, there is an urgent need to develop novel methods for the diagnosis and classification of pancreatic cancer using AI. Various large and robust models with high accuracy are being developed every day. 

One of the main applications of AI in pancreatic cancer diagnosis is the analysis of imaging data, such as computed tomography (CT), magnetic resonance imaging (MRI), positron emission tomography (PET), and endoscopic ultrasound (EUS). AI can help to detect pancreatic lesions, measure their size and shape, assess their malignancy and invasiveness, and predict their response to therapy. For example, a deep learning model based on convolutional neural networks (CNNs) was developed to automatically segment pancreatic tumors from CT images and classify them into resectable or unresectable categories with high accuracy. Another deep learning model based on recurrent neural networks (RNNs) was proposed to analyze EUS images and differentiate between benign and malignant pancreatic cysts with high sensitivity and specificity.

Another application of AI in pancreatic cancer diagnosis is the analysis of genomic data, such as DNA sequencing, RNA sequencing, microarray, and methylation data. AI can help to identify genetic mutations, gene expression patterns, epigenetic modifications, and molecular subtypes of pancreatic cancer that are associated with prognosis and treatment response. For example, a machine learning model based on support vector machines (SVMs) was developed to classify pancreatic cancer into four molecular subtypes (basal-like, classical, quasi-mesenchymal, and exocrine-like) based on gene expression data. Another machine learning model based on random forests (RFs) was proposed to predict the survival of pancreatic cancer patients based on DNA methylation data.

AI can also integrate multiple types of data to provide a comprehensive diagnosis and classification of pancreatic cancer. For example, a deep learning model based on multimodal autoencoders (MAEs) was developed to fuse imaging data (CT and PET) and genomic data (gene expression) to classify pancreatic cancer into resectable or unresectable categories. 

Several AI algorithms have demonstrated promising performance in real-world clinical settings, particularly in applications like medical imaging and diagnostics [104,105,106,107,108,109]. They show potential for enhancing healthcare outcomes by rapidly analyzing extensive medical data for early disease detection. However, challenges remain, including the need for rigorous validation, adaptation to diverse patient populations, and seamless integration into healthcare workflows that we further discuss in a later part of this article. 

### 3.3. Treatment Planning and Monitoring

As the complexity of the disease necessitates a multidisciplinary approach, AI emerges as a transformative tool in cancer management [3]. The inherent heterogeneity of pancreatic cancer demands personalized treatment approaches tailored to individual patients. By integrating patient-specific data, including clinical history, genomic profiles, and imaging results, AI can assist oncologists in formulating customized treatment plans [110,111,112,113]. These plans encompass a spectrum of therapeutic modalities, such as surgery, chemotherapy, radiation therapy, and immunotherapy, optimizing the chances of treatment success while minimizing potential adverse effects [114,115,116,117,118,119,120,121]. AI-enabled predictive modeling takes into account various patient-related factors, genetic markers, and tumor characteristics to forecast an individual’s response to specific treatments. These models provide valuable insights to clinicians, aiding in informed treatment decisions and optimizing therapeutic strategies for improved patient outcomes.

By analyzing real-time patient data, including biomarker levels and imaging results, AI algorithms can detect treatment efficacy or disease progression promptly. The early identification of treatment failure allows timely modifications to the treatment regimen, enhancing the chances of successful disease management.

### 3.4. Biomarker Discovery

Biomarkers are the measurable indicators of biological processes or conditions that can be used for the diagnosis, prognosis, or monitoring of diseases [122,123]. Biomarkers can be derived from various sources, such as blood, urine, saliva, tissue, or genetic material [4,124,125,126,127]. For pancreatic cancer, biomarkers can help detect the disease at early stages, differentiate between benign and malignant lesions, classify the tumor subtype, predict the response to therapy, and monitor disease progression or recurrence [128].

However, finding reliable and specific biomarkers for pancreatic cancer is challenging due to the heterogeneity and complexity of the disease, the lack of adequate samples, and the interference of confounding factors [128]. AI can help overcome these challenges by applying advanced computational methods to analyze large and diverse datasets of biomolecular information, such as genomics, proteomics, metabolomics, or microbiomics. AI can also integrate multiple types of data from the pancreas to identify novel biomarkers or biomarker signatures that have higher sensitivity and specificity than single biomarkers [129,130,131,132]. A deep learning model based on multimodal neural networks (MNNs) was proposed to combine imaging data (WSI), gene expression data, clinical data (age, gender, tumor location), and biomarker data (mi-RNA) to forcast the survival of pancreatic cancer patients [133].

### 3.5. Contrast-Enhanced Ultrasound 

More recently, AI based on contrast-enhanced ultrasound has been emerging to characterize pancreatic cancer. For instance, a 2022 study developed a deep-learning radiomics model based on contrast-enhanced ultrasound images to aid radiologists in the diagnosis of pancreatic ductal adenocarcinoma [134]. The study included 558 patients, and their model achieved an AUC above 0.95 in their training, internal validation, and two external validation cohorts. Improvements in the diagnosis of pancreatic cancer have significant clinical implications as they may avoid invasive diagnostic interventions. AI has also improved the capabilities in predicting treatment responses in pancreatic cancer as well as the cancer’s aggressiveness. For instance, in a 2023 study of 38 patients, investigators were able to utilize the deep learning of contrast-enhanced ultrasound videos to predict the efficacy of neoadjuvant chemotherapy for pancreatic cancer, achieving AUCs of above 0.89 in two CNN models [135]. In a 2022 study of 104 patients, investigators were able to develop and validate a nomogram that included both clinical factors (e.g., tumor size, arterial enhancement level, deep learning predictive probability) and deep learning contrast-enhanced ultrasound to predict the aggressiveness of patients’ pancreatic neuroendocrine neoplasms preoperatively. Their combined nomogram was able to better predict the aggressiveness of the tumors than the clinical model (AUC, 0.97 vs. 0.87, *p* = 0.009). AI can also be used to automatically segment pancreatic masses. A 2021 study was able to use deep learning to automatically segment pancreatic tumors on contrast-enhanced endoscopic ultrasound with a decent concordance rate [136]. Finally, AI has also been used to help distinguish between pancreatic cancer from non-cancerous masses using contrast-enhanced ultrasound. A 2023 prospective trial demonstrated a deep learning-based system that was able to diagnose pancreatic masses significantly better than that of endoscopists [137]. The model was also able to improve first-pass diagnostic yield when used to guide fine needle aspiration.

### 3.6. Healthcare Workflows

Not only has AI been able to improve operational efficiency, but it also has several practical implications in healthcare workflows [138,139,140,141,142]. Some examples include (a) the area of diagnostic imaging by having a faster and more accurate analysis of images which aids in early detection and diagnosis; (b) clinical decision making support by improvement and adhering to best practice standards and providing personalized treatment plans; (c) utilizing NLP in electronic health records; and (d) drug discovery, cost reduction, and development of targeted therapies.

The challenges faced during implementation are to ensure that data used by AI systems are accurate, complete, and interoperable across different healthcare systems. In addition, other challenges include navigating complex regulatory frameworks, gaining acceptance from healthcare professionals who may be skeptical about relying on AI recommendations for patient care, balancing the costs associated with implementing AI solutions against the expected benefits, and demonstrating a clear ROI. These challenges require collaboration among healthcare professionals, technology developers, policymakers, and regulatory bodies to create a supportive and secure environment for the integration of AI in healthcare workflows.

## 4. Limitations and Challenges

Currently, surgery and chemotherapy are the main treatment options for pancreatic cancer. However, only a small portion of patients (15–20%) are candidates for surgical resection [143]. In those who undergo surgery, nearly 75% of patients develop recurrence within 2 years, suggesting the presence of micro-metastatic disease [144]. Regardless of whether the patient undergoes surgical resection, most patients with pancreatic cancer receive systemic chemotherapy. Two combination regimens serve as first-line treatment options known as FOLFIRNOX and nab-paclitaxel [118,145]. There are currently no standardized treatment algorithms for second-line regimens and treatment decisions. Patients who undergo surgery are usually treated with adjuvant chemotherapy in the postoperative period with one of the modified combined regimens [146,147]. Neo-adjuvant therapy in the pre-operative period is also being adopted to shrink the malignancy potentially and has shown improved overall survival in patients who receive this treatment [148]. These advances in the treatment options for pancreatic cancer are promising but rely on the early detection of pancreatic cancer, which is complicated by its insidious progression and propensity to metastasize prior to detection [143]. 

Additionally, treatment and prevention are further complicated by the lack of a single or few attributable causes contributing to the development of pancreatic cancer in most patients. A small burden of cases can be attributed to specific risk factors such as germline mutations, chronic pancreatitis, and mucinous cystic lesions [149]. Currently, two pathways for tumor progression have been recognized, each with distinct characteristics. The first, more frequent, is Pancreatic Intraepithelial Neoplasia (PanINs), which are microscopic pre-lesions that cannot currently be identified through imaging [150]. The second, less frequent, are Intraductal Papillary Mucinous Neoplasms (IPMNs), which can be identified through imaging [151]. Complex and varying tumor biology contributes to an additional layer of complexity in identifying treatment targets. Recent advances in the treatment landscape have focused on targeting tumor genetic and immunologic characteristics [152,153], changes in tumor metabolism [154], and tumor microenvironment [155]. Further investigation into the mechanisms of tumor development and the efficacy of evolving treatment options will be important to making an impact on current treatment models of pancreatic cancer. 

AI can serve as a powerful tool in the advancement of pancreatic cancer diagnosis, management, and prognosis, particularly in identifying tumors earlier in disease progression. Despite the many applications and advantages of AI in pancreatic cancer, multiple limitations pose challenges that must be addressed as the field grows. One is the lack of a standardized approach to treatment and diagnosis. Other challenges include a lack of robust and high-quality data, transparency and reproducibility of findings, and ethical considerations, including biases in algorithms.

The lack of large, centralized datasets that can be used to build and test algorithms poses a barrier to developing comprehensive models. Currently, there is only one major effort in addressing this through the NIH-NCI-sponsored EDRN project for pancreatic cancer [19]. Studies that have used smaller available datasets have not accounted for suboptimal image quality and factors that make images unsuitable for AI, such as post-treatment status and the presence of biliary stents [156,157]. These gaps in the quality of the data used to develop models may result in errors and biases that limit their applications in clinical medicine. 

Furthermore, AI algorithms have been previously referred to as “Black boxes” due to their lack of transparency and interpretability [19,158,159]. The opacity of the code used to build AI models and the hidden level of complexity make it difficult to reproduce results in an independent manner [160,161,162]. General descriptions of the code used to build models do not provide enough information to reproduce most findings. The lack of easy interpretation of these AI models and prospective studies assessing AI-based tools has increased the hesitancy of adaptation into clinical practice. Without transparency and interpretation, clinicians are not able to critically interrogate the output of these models, putting an incredible amount of faith in the accuracy of the model [163]. Improving reproducibility and interpretability will be crucial challenges to overcome prior to the clinical adaptation of AI models in pancreatic cancer. Nonetheless, there have been key initiatives and techniques implemented that are crucial for gaining the trust of healthcare professional and patients [164,165,166,167]. Research are actively working on developing explainable AI techniques such as generating feature importance scores, visualizations, and natural language explanations to convey the reasoning behind AI predictions. More alignment of AI outputs with existing clinical guidelines is needed to simplify the understanding of how AI recommendations align with established medical practices. Using interactive interfaces, promoting transparency in the development of AI algorithms, incorporating ethical considerations into the development of AI systems, and encouraging collaboration between AI developers and medical experts ensures that AI systems are designed with a deep understanding of the medical context and align with the decision-making processes of healthcare professionals.

Lastly, a few ethical concerns should be considered when discussing the implementation of AI in pancreatic cancer. Datasets used to build models tend to lack data from underrepresented groups such as women and minorities, leading to biased models that may not be applicable to the diverse patient population seen clinically [168]. Implementing these skewed algorithms can increase disparities in health outcomes between groups rather than improving outcomes [14,169], particularly because models tend to perform best on data that are most like the data they were trained with. Improving the diversity in patient data used to train models and validating models across various populations could mitigate this challenge and provide models that are more generalizable to a heterogeneous patient population. Additionally, the creation and use of large datasets needed to create AI models pose the challenging questions of data ownership and patient privacy, particularly in reference to medical imaging [163,170]. At the same time, the integration of AI systems in medical practices raises questions about the security and confidentiality of sensitive patient data [171,172]. Ensuring robust data protection mechanisms is imperative to prevent unauthorized access and potential misuse of personal health information [173,174,175]. Additionally, ethical challenges encompass issues such as algorithmic bias, transparency, and accountability [176,177,178]. Addressing these challenges requires the establishment of ethical guidelines and regulatory frameworks that prioritize fairness, transparency, and the responsible use of AI technologies [179,180]. Striking a balance between innovation and ethical considerations is essential to foster public trust and promote the responsible adoption of AI in healthcare, ultimately ensuring that advancements in technology benefit patients without compromising their privacy or perpetuating existing healthcare disparities.

It is crucial to consider the many challenges and limitations that exist in the clinical utility of AI in pancreatic cancer and medicine at large. Improving the generalizability of models through high-quality datasets, improving transparency of the methods used to create models, addressing biases in algorithms, and posing solutions to patient privacy concerns are the immediate hurdles that must be addressed prior to AI transforming patient care. 

## 5. Future Directions

Emerging technologies in this space have practical implications for reducing morbidity and mortality with promising new research coming out, such as the recent work that was able to use radiomics-based machine learning models to detect pancreatic ductal adenocarcinoma at the prediagnostic stage with substantial lead time [181].

This section outlines several significant areas where further research and development are required to enhance the use of AI in pancreatic cancer.

**Multi-omics data integration.** Pancreatic cancer is a complicated disease with molecular heterogeneity. Integrating multi-omics data, such as genomes, transcriptomics, proteomics, and metabolomics, can offer a complete picture of the disease pathology. Future research should concentrate on building artificial intelligence algorithms capable of assessing and combining these disparate datasets in order to uncover strong molecular signatures, biomarkers, and therapeutic targets for pancreatic cancer.

**Improved imaging analysis.** To increase the quality and efficiency of radiological examinations, AI algorithms can aid in automated image analysis, segmentation, and feature extraction. Future research should attempt to enhance and verify AI models particularly intended for pancreatic cancer imaging analysis, allowing for better disease identification, staging, and monitoring.

**Predictive modeling and risk stratification:** Artificial intelligence systems have shown potential in forecasting medical outcomes and categorizing patients into risk categories based on clinical and genomic markers. Future research should concentrate on establishing strong prediction models that include clinical, genetic, and imaging data to identify individuals who are at high risk of disease progression, recurrence, or therapy resistance. Such models can help with tailored therapy selection and patient care techniques.

**Real-time decision support systems:** AI-driven decision support systems can aid doctors in real-time treatment planning and monitoring by offering evidence-based suggestions. To guide treatment decisions, these systems can incorporate patient-specific data, clinical guidelines, and relevant scientific research. Future research should concentrate on building user-friendly, interpretable, and scalable AI algorithms that can be smoothly incorporated into clinical processes to help healthcare practitioners make educated decisions for patients.

**Data sharing and collaboration:** The availability of high-quality, diversified, and well-annotated datasets is critical to the effectiveness of AI in pancreatic cancer research. It is critical to promote data sharing and collaboration across research institutes and clinical facilities in order to advance AI applications in pancreatic cancer. Future initiatives should concentrate on developing standardized data-gathering techniques, ethical data-sharing frameworks, and safe data integration and analysis platforms.

**Clinical validation and implementation:** In order to convert AI research into clinical practice, robust validation studies in pancreatic cancer are required to establish the clinical efficacy, safety, and cost-effectiveness of AI-based methods. Large-scale prospective studies should be conducted in the future to evaluate the performance of AI algorithms in real-world healthcare situations. Furthermore, regulatory and ethical factors such as privacy protection, informed consent, and algorithm transparency must be addressed to enable responsible and fair AI technology implementation in healthcare.

## 6. Conclusions

In conclusion, this comprehensive review highlights the immense potential of artificial intelligence in advancing the management of pancreatic cancer and improving patient outcomes. The integration of machine learning algorithms and AI techniques has shown significant progress in various aspects of pancreatic cancer, including early detection, accurate diagnosis, treatment selection, and prognosis prediction. By leveraging the power of AI and harnessing vast amounts of data, healthcare professionals can make more informed decisions and provide personalized care to patients. However, it is essential to address challenges related to data availability, model interpretability, and ethical considerations to ensure the responsible and effective implementation of AI in pancreatic cancer care. With continued research and collaboration, AI has the potential to revolutionize the field and contribute to better outcomes for individuals affected by this challenging disease.

## Figures and Tables

**Figure 1 diagnostics-14-00174-f001:**
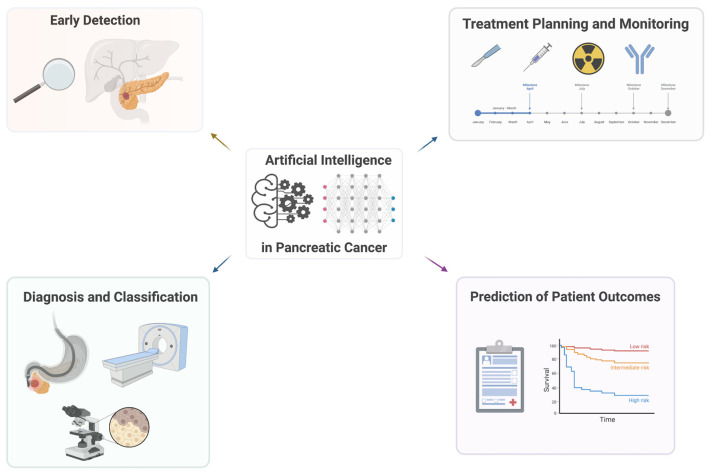
Summary of the applications of artificial intelligence in pancreatic cancer.

**Figure 2 diagnostics-14-00174-f002:**
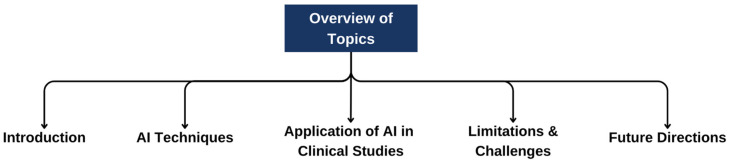
Overview of the topics covered in this article.

**Figure 3 diagnostics-14-00174-f003:**
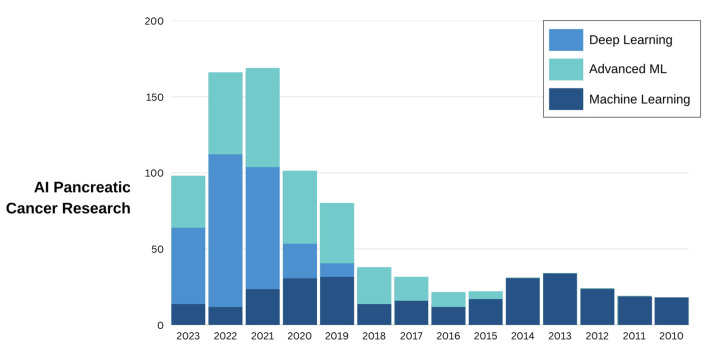
Timeline of AI in pancreatic cancer. Machine learning is categorized here as vanilla algorithms that learn from data and make decisions or predictions, like linear models and support vector machines. Advanced machine learning incorporates complex algorithms and methods such as ensemble techniques and reinforcement learning. Deep learning methods focus on algorithms inspired by the structure and function of the brain, specifically deep neural networks.

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
