# Peer review of "From Machine Learning to Patient Outcomes: A Comprehensive Review of AI in Pancreatic Cancer"

_diagnostics, 2024, doi:10.3390/diagnostics14020174_

Round 1

Reviewer 1 Report

Comments and Suggestions for Authors

This article provides a comprehensive overview of the application of AI technology in pancreatic cancer diagnosis, encompassing multiple AI domains, namely machine learning, deep learning, natural language processing, computer vision, and transfer learning. However, as a review article, this paper still lacks a more extensive discussion. For instance:

The article does not mention the latest research frontier in AI, which is the application of large models in medical diagnostics.

The paper only briefly outlines some AI diagnostic methods, and in my opinion, it would benefit from the inclusion of additional experimental data for analysis.

Comments on the Quality of English Language

Moderate editing of English language required

Author Response

Reviewer #1:

Reviewer #2 Comment #1: This article provides a comprehensive overview of the application of AI technology in pancreatic cancer diagnosis, encompassing multiple AI domains, namely machine learning, deep learning, natural language processing, computer vision, and transfer learning. However, as a review article, this paper still lacks a more extensive discussion. For instance:

The article does not mention the latest research frontier in AI, which is the application of large models in medical diagnostics.

The paper only briefly outlines some AI diagnostic methods, and in my opinion, it would benefit from the inclusion of additional experimental data for analysis.

Response: We thank the reviewer for their useful suggestion to include a more extensive discussion. We have added further relevant discussion in the first paragraph of section “3. Application of AI in Clinical Studies.” Furthermore, we have added further discussion throughout the manuscript to address Reviewer #2’s suggestion, which we hope will also further help address the extensive discussion suggestion.

Reviewer 2 Report

Comments and Suggestions for Authors

1. It would be helpful to include specific references or citations to the sources of data and studies mentioned in the text, such as the statistics on pancreatic cancer incidence and the advancements in imaging technology. Providing sources would enhance the credibility of the information presented.

2. A thorough proofreading of the document is suggested.

3. While the abstract provides a good introduction to the topic, it could benefit from a more in-depth discussion of the various applications of AI in pancreatic cancer care. For example, specific AI techniques and models used for diagnostics and personalized medicine could be discussed in more detail.

4. The abstract touches on data privacy and ethical considerations briefly. Expanding on this aspect and discussing the ethical challenges and solutions related to AI in healthcare would provide a more comprehensive overview.

5. It is mentioned that AI can improve diagnosis, but it would be valuable to elaborate on specific examples or studies that demonstrate the clinical validation of AI systems in pancreatic cancer care. How well have AI algorithms performed in real-world clinical settings?

6. The importance of interpretability in AI models for healthcare is briefly mentioned. Further discussion on efforts to make AI systems more interpretable and transparent, especially in the context of medical decision-making, would be beneficial.

7. The paper mentions operational efficiency improvements through AI, but it would be interesting to explore practical examples of how AI is currently being integrated into healthcare workflows and the challenges faced during implementation.

8. Including a section on the potential future directions of AI in pancreatic cancer care, such as upcoming research trends or emerging technologies, would provide a forward-looking perspective.

9. The paper could benefit from a more robust conclusion that summarizes the key takeaways and emphasizes the potential transformative impact of AI in improving patient outcomes in pancreatic cancer care.

10. The text is generally well-written and clear. However, ensuring consistent terminology and avoiding repetition can further enhance readability.

Comments on the Quality of English Language

A thorough proofreading of the document is suggested.

Author Response

Reviewer #2:

Reviewer #2 Comment #1: It would be helpful to include specific references or citations to the sources of data and studies mentioned in the text, such as the statistics on pancreatic cancer incidence and the advancements in imaging technology. Providing sources would enhance the credibility of the information presented.

Response: We thank the reviewer for their suggestions. Additional references have been added to throughout the revisions to enhance the credibility of the information presented.

Reviewer #2 Comment #2: A thorough proofreading of the document is suggested.

Response: We have done a thorough proofreading of the manuscript on the revised version. 

Reviewer #2 Comment #3: While the abstract provides a good introduction to the topic, it could benefit from a more in-depth discussion of the various applications of AI in pancreatic cancer care. For example, specific AI techniques and models used for diagnostics and personalized medicine could be discussed in more detail.

Response: We thank the reviewer for this useful suggestion. We have added extensive discussion on this in the first paragraph of the “2. AI Techniques” section. We kept the abstract short in order to not exceed the word count requirement but hope this better addresses the revision suggestion.

Reviewer #2 Comment #4: The abstract touches on data privacy and ethical considerations briefly. Expanding on this aspect and discussing the ethical challenges and solutions related to AI in healthcare would provide a more comprehensive overview.

Response: We thank the reviewer for their insightful suggestion. We have expanded this more in the 4. Limitations & Challenges section, adding a paragraph on the ethical challenges and solutions related to AI in healthcare. Providing a more comprehensive overview of the current state of AI.

Reviewer #2 Comment #5: It is mentioned that AI can improve diagnosis, but it would be valuable to elaborate on specific examples or studies that demonstrate the clinical validation of AI systems in pancreatic cancer care. How well have AI algorithms performed in real-world clinical settings?

Response: We appreciate the reviewer for this helpful suggestion. We have added a paragraph in the end of the “3.2. Diagnosis & Classification” section talking about how AI can translate into real-world clinical settings and citing multiple relevant cases.

Reviewer #2 Comment #6: The importance of interpretability in AI models for healthcare is briefly mentioned. Further discussion on efforts to make AI systems more interpretable and transparent, especially in the context of medical decision-making, would be beneficial.

Response: We thank the reviewer for this useful suggestion. We have added relevant discussion to the 5th paragraph of our “4. Limitations & Challenges” section.

Reviewer #2 Comment #7: The paper mentions operational efficiency improvements through AI, but it would be interesting to explore practical examples of how AI is currently being integrated into healthcare workflows and the challenges faced during implementation.

Response: We appreciate the reviewer’s excellent suggestion. We have added a new section to our manuscript titled “3.5 Healthcare workflows.”

Reviewer #2 Comment #8: Including a section on the potential future directions of AI in pancreatic cancer care, such as upcoming research trends or emerging technologies, would provide a forward-looking perspective.

Response: We appreciate the reviewer for this suggestion. Our section “5. Future Directions” details the various areas of potential future directions of AI in pancreatic cancer with a forward-looking perspective. We have expanded the first paragraph of this section to emphasize emerging technologies and have added a recent citation of impactful work.  

Reviewer #2 Comment #9: The paper could benefit from a more robust conclusion that summarizes the key takeaways and emphasizes the potential transformative impact of AI in improving patient outcomes in pancreatic cancer care.

Response: We thank the reviewer for this great suggestion and have added more summary of the key takeaways to our conclusion.

Reviewer #2 Comment #10: The text is generally well-written and clear. However, ensuring consistent terminology and avoiding repetition can further enhance readability.

Response: We appreciate this suggestion and have been careful to implement consistent terminology and avoid repetition in the revised text.

Round 2

Reviewer 1 Report

Comments and Suggestions for Authors

(1) The arrangement of the manuscript in left-end alignment is uncomfortable, and all alignments should be unified into two-end alignment.

(2) The plain text description makes this review paper unreadable, and the mind map and domain classification and related picture display are necessary in my opinion.

Comments on the Quality of English Language

Minor editing of English language required

Author Response

We thank the reviewer for their suggestions and useful insights.

  1. Unfortunately, since we have formatted the manuscript in the MDPI Diagnostics journal-specific format, we can not change the formatting and alignment of the text.

  2. We have made a new figure (Figure 2), which we added to the introduction section to provide a more visual overview of the manuscript.